# Assessment of Morpho-Physiological, Biochemical and Antioxidant Responses of Tomato Landraces to Salinity Stress

**DOI:** 10.3390/plants10040696

**Published:** 2021-04-05

**Authors:** Reem H. Alzahib, Hussein M. Migdadi, Abdullah A. Al Ghamdi, Mona S. Alwahibi, Abdullah A. Ibrahim, Wadei A. Al-Selwey

**Affiliations:** 1Department of Botany and Microbiology, College of Sciences, King Saud University, Riyadh 11451, Saudi Arabia; alzaheb77@yahoo.com (R.H.A.); aaalgham@yahoo.com (A.A.A.G.); malwhibi@ksu.edu.sa (M.S.A.); 2College of Food and Agriculture Sciences, King Saud University, P.O. Box 2460, Riyadh 11451, Saudi Arabia; adrahim@ksu.edu.sa (A.A.I.); wadeiahmed@gmail.com (W.A.A.-S.)

**Keywords:** *Solanum lycopersicum*, enzyme activity, ROS, SOD, proline, K^+^/Na^+^

## Abstract

Understanding salt tolerance in tomato (*Solanum lycopersicum* L.) landraces will facilitate their use in genetic improvement. The study assessed the morpho-physiological variability of Hail tomato landraces in response to different salinity levels at seedling stages and recommended a tomato salt-tolerant landrace for future breeding programs. Three tomato landraces, Hail 548, Hail 747, and Hail 1072 were tested under three salinity levels: 75, 150, and 300 mM NaCl. Salinity stress reduced shoots’ fresh and dry weight by 71% and 72%, and roots were 86.5% and 78.6%, respectively. There was 22% reduced chlorophyll content, carotene content by 18.6%, and anthocyanin by 41.1%. Proline content increased for stressed treatments. The 300 mM NaCl treatment recorded the most proline content increases (67.37 mg/g fresh weight), with a percent increase in proline reaching 61.67% in Hail 747. Superoxide dismutase (SOD) activity decreased by 65% in Hail 548, while it relatively increased in Hail 747 and Hail 1072 treated with 300 mM NaCl. Catalase (CAT) activity was enhanced by salt stress in Hail 548 and recorded 7.6%, increasing at 75 and 5.1% at 300 mM NaCl. It revealed a reduction in malondialdehyde (MDA) at the 300 mM NaCl concentration in both Hail 548 and Hail 1072 landraces. Increasing salt concentrations showed a reduction in transpiration rate of 70.55%, 7.13% in stomatal conductance, and 72.34% in photosynthetic rate. K+/Na+ ratios decreased from 56% for 75 mM NaCl to 85% for 300 mM NaCl treatments in all genotypes. The response to salt stress in landraces involved some modifications in morphology, physiology, and metabolism. The landrace Hail 548 may have better protection against salt stress and observed protection against reactive oxygen species (ROS) by increasing enzymatic “antioxidants” activity under salt stress.

## 1. Introduction

Salinity is a significant abiotic stress, affecting plant growth and productivity during all plant developmental stages. Worldwide, 800 million ha of land and 32 million ha of agricultural land are salt-affected [1], representing about 20% of cultivated land and 33% of irrigated land degraded in the world [2]. The shortage of good quality water is becoming an important issue. For this reason, the use of saline water is becoming essential and should receive immediate consideration. To enhance productivity, improving the salt tolerance of crop plants can make marginal areas productive [3].

Tomato is listed in the top of 20 commodities grown worldwide and, in terms of vegetable production, is second only to that of potato with a worldwide production of over 182 million tons covering over 4.7 million ha [4]. Tomato grows in diverse climatic conditions, but its optimal cultivation areas are found in warm and somewhat dry regions, such as Mediterranean countries [5]. In these areas, water and salinity stresses are common environmental factors that reduce crop yields. Several investigators have reported that soil salinity reduces plant growth and the productivity of many crops, including most vegetable crops, which present low soil salinity tolerance. Salinity reduces plant growth because of osmosis; a reduction in water availability to plants and because of ionic effects; specific ion toxicity and mineral nutrient deficiencies on soil solution [6]. An increase in salt content in the growing environment increases salt stress on plant growth. Salinity results in reduced yield because of the reduction in photosynthesis efficiency, chlorophyll, total protein, biomass, stomata closure, and increasing oxidative stress [7,8,9]. The salinity response varies from plant to plant, the stage of growth, and the salinity level. For example, in cereals crops such as wheat, rice, and maize, the biomass decreased at 100 to 150 mM NaCl level, while sunflower and tomato weight decreased at 50 mM NaCl. In citrus trees, biomass decreased at 100 mM NaCl; in contrast, palm showed less susceptibility to severe stress. However, at 200–300 mM salt (NaCl), all plants showed adverse effects [10].

Moreover, Bashir et al. [11] reported that after four weeks of exposure of olive genotypes to salt stress, all genotypes showed typical toxicity symptoms, such as increased chlorosis and decreased chlorophyll content, which reached 57% at the highest NaCl concentration (200 mM), reduced relative growth rate, lipid peroxidation by enhanced malondialdehyde (MDA) content. The proline accumulation in the salt-tolerant cultivar was significantly higher (36%) than the salt-sensitive one, and a lower accumulation of protein in the shoots was reported.

Most crop plants, including tomatoes, are sensitive to salinity throughout ontogeny. The response to salt stress involves modifications in morphology, physiology, and metabolism. Understanding plant physiology, genetics, and molecular biology are essential for breeding new cultivars to grow under saline conditions with similar crop productivity under normal conditions. Plants can adapt to salinity stress through various mechanisms, including osmotic regulation, ion uptake and transport, antioxidant metabolism, hormone metabolism, and stress signaling. Increasing salinity level associated with the reduction in tomato growth [12,13,14], reduction in chlorophyll content, and photosynthesis-related traits [13,15,16,17], enhancing proline accumulation [18,19], antioxidant metabolism activities [19,20,21,22,23], and K^+^/Na^+^ ratio [13,24].

Unlike tomato hybrids, tomato landraces are less sensitive to environmental stresses and are grown under low inputs, still grown in small farms because of consumer quality and particular demand. These landraces are valuable sources of genetic characteristics for plant breeders’ interest in breeding crop improvement programs. This study assessed the morpho-physiological variability of tomato landraces grown under different salinity levels at seedling stages and recommend the promising tomato salt-tolerant landrace for future breeding programs.

## 2. Results

### 2.1. The Effect of Salinity on Some Growth Parameters

Salinity stress induced by NaCl treatments influenced and reduced the leaf area, plant height, shoot and root fresh and dry weights compared with a control, particularly at high concentrations (300 Mm) of NaCl (Table 1). The interaction between landraces and salinity was only significant in leaf area traits, showing that they were more influenced by salt stress treatment. The relative percentage of change (decrease or increase) in growth parameters is presented in Appendix A**.** The response of landraces to salinity was also different under salt treatments. The Hail 548 landrace showed less reduction in these traits than other landraces Hail 747 and ‘Hail 1072 at a significant level. For example, compared to the control treatment, the leaf area reduced with a range from 1.96% in Hail 548 to over 45% in both Hail 747 and Hail 1072 landraces. The response of shoot and root weight was also treatment and landrace dependent. Hail 548 landrace showed less effects and a reduction in shoot fresh and dry weight (54.44% and 61.29%) than Hail 1072 83.19% and 84.22% for shoot fresh and dry weight, respectively, at the highest salt stress treatment, 300 mM NaCl. Root fresh weight recorded a reduction of 77.48% in Hail 548 and 93.89% in Hail 1072, and dry weight was 66.67% and 91.8% in Hail 548 and Hail 747, respectively, at 300 mM NaCl treatment.

### 2.2. The Effect of Salinity on Leaf Pigments, Free Proline, Protein Content, Soluble Sugars

Neither the landraces nor salt treatments showed a significant effect difference at the 5% level, including chlorophyll-a, chlorophyll-b, total chlorophyll, and carotenoids contents (Table 2). The percentage of changes compared with the control in chlorophyll and carotene pigments in landraces are present in Appendix A**.** The Hail 747 landrace showed an enhancement in leaf pigments, which reached over 41%, 37%, 40%, and 44.7% for chlorophyll-a, chlorophyll-b, total chlorophyll, and carotenoid contents, respectively. However, both landraces Hail548 and Hail1072 showed a reduction in all pigments.

The most increases in proline content were associated with the highest salt level treatment. The values ranged from 2.53 at control to 67.37 mg/g fresh weight at 300 mM NaCl treatments. Hail 747 recorded the highest proline accumulation, and Hail548 recorded the lowest content. The percent increase reached 61.67% in Hail 747. Although landraces showed significant responses, salt treatments did not significantly reduce the protein content (Table 3). The Hail 548 landrace significantly accumulated more protein in leaves compared with both Hail 747 and Hail 1072. However, the percentage of reduction-in-protein content varied among genotypes and salt concentrations. Hail 747 showed the lowest reduction in protein content Appendix A. A significant difference among landraces was recorded for soluble sugars. Hail 548 recorded the highest soluble sugar mean value, 13.826 mg/g leaf dry weight, and Hail 1072 recorded the lowest, 9.453 mg/g leaf dry weight. Although salt stress showed non-significant effects on soluble sugar traits, increasing salt level was associated with decreased soluble sugars Appendix A. Except for landrace Hail 548, in both landraces (Hail 747 and Hail 1072), soluble sugars were enhanced at the low salt concentration treatment of 75 mM NaCl.

### 2.3. The Effect of Salinity on Some Antioxidant Parameters and MDA Content

The activity of superoxide dismutase (SOD) increased in both Hail 747 and Hail 1072 landraces with an increase in salinity. However, the catalase (CAT) activity showed a somewhat fluctuating response; activity increased under saline conditions gradually by increasing the salt concentration in Hail 548 and at the 300 mM NaCl in Hail 747. CAT activity decreased by increasing salinity in the Hail 1072 landrace. (Table 4). A non-significant difference among landraces, salinity treatments, and their interaction was found in the MDA concentration (an indicator of lipid peroxidation). However, an increase in salinity resulted in a rise in MDA concentration at 150 and 300 mM NaCl in the Hail 747 landrace, but the reduction in MDA at higher concentrations of salinity was revealed in both the Hail 548 and Hail 1072 landraces. The reduction in MDA was 9% in Hail 548 and reached 38.46% in Hail 1072 at 300 mM NaCl Appendix A.

### 2.4. The Effect of Salinity on Some Photosynthetic Parameters and Relative Water Content

Increasing salt concentrations significantly reduced the photosynthesis rate and related traits; however, increased relative water content in Hail 548 and Hail 1072 at both 75 and 150 mM NaCl (Table 5). Compared to the control treatment, the most significant reduction in transpiration rate occurred at the concentration of 300 mM NaCl by 70.55% on average, and Hail 548 reported the highest reduction (88.6%) at 300 mM NaCl. Regardless of the salt concentration, Hail 548 revealed lower stomatal conductance values, while Hail 1072 revealed high conductance at high salt concentrations of 150 and 300 mM NaCl. The photosynthetic rate significantly decreased with the increasing concentration of NaCl, and landraces also revealed differences. According to increasing salt ingredients, the reduction percentage ranged from 46.74% to 72.34% for Hail 548 and Hail 1072, respectively. Significant differences were recorded among landraces and salt treatments and their interaction for relative water content. The percentage increased in Hail 548 at 75 mM NaCl was 17.25% and 20.5% in Hail 1072 (Appendix A).

### 2.5. The Effect of Salinity on Sodium, Potassium, and Na^+^/K^+^ Ratio

Salinity induced a higher Na^+^ uptake in all landraces, along with a lowered K^+^ content in both Hail 747 and Hail 1072, while it increased the Hail 548 landrace compared to the unstressed control plants (Table 6). In Hail 548, the content increased eleven-fold at 300 mM NaCl, while it was two-fold in Hail 1072 and 6.7 folds in Hail 747. The highest percentage of increases in leaves were 988% in Hail 548 at 300 mM NaCl, 690%, and 106% at 150 mM NaCl in Hail 747 and Hail 1072 leaves. K^+^ accumulation increased in Hail 548 by 1.4-fold compared to the control at the highest NaCl concentration, while a gradient reduction in both landraces Hail 747 and Hail 1072 by 0.7- and 0.6-fold, respectively. According to the salinity level, the percentage of K^+^ increased or decreased was a 39.39% increase in Hail 548, whereas there was a 34.88% and 37.48% reduction in Hail 747 and Hail 1072, respectively (Appendix A). The data shown in Table 6 show that the K^+^/Na^+^ ratios decreased in all genotypes as the salinity level increased. Although landraces showed no significant differences, the K/Na ratios of the Hail 747 revealed the highest ratio (8.59) and were followed by Hail 548 and Hail 1072 with values 6.07 and 5.0, respectively. The percentage of reduction in the K/Na ratio ranged from 47.72% to 87.1% in Hail 548, 69.0% to 90.14% in Hail 747, and 37.0% to 68.8% Hail 1072 Appendix A. Simultaneously, the percent reduction in K/Na ranged from 56% for 75 mM NaCl to 85% for 300 mM NaCl treatments.

## 3. Discussion

### 3.1. The Effect of Salinity on Some Growth Parameters

Salinity is a significant constraint limiting agricultural crop productivity in the world. However, plant species and cultivars differ significantly in their response to salinity. The response of tomato growth to salinity stress is genotype and concentration dependent. The effect of salinity is critical to any growth and development stage of the plant. Several reports showed that there is a linear correlation between the reduction in plant growth with increasing salinity levels. Genotypes can be selected as salt-tolerant at the early development stage, based on the severity of symptoms and their dry matter production [12,25,26]. However, Raza et al. [13] found that the root length, fresh and dry root weight, root/shoot ratio, and shoot Na^+^ concentration significantly increased with increasing salinity. The plant that tolerates salinity at an early growth (seedling stage) may improve tolerance at other growth stages [25]. Hence, selecting salt-tolerant plants at a seedling stage has been practiced in many crops, including tomato. The reduction in root and shoot fresh and dry weight was discussed and correlated with many reasons.

Ion toxicity, osmotic stress, and nutritional deficiency lead to oxidative stress [27], harm the metabolic imbalance in plants, and inhibit and retard growth. The reduction in chlorophyll content and photosynthesis rate because of salinity may bring other abnormal changes in the plant body’s functioning, ultimately reducing overall growth and productivity [25]. Hand et al. [14] reported that the salt-sensitive “genotype’s growth parameters” reduction results from several physiological responses. They include the modification of ion balance, mineral nutrition, stomatal behavior, and photosynthetic efficiency. Many reports showed NaCl reduces the plant’s ability to take up water and leads to slow growth; then excessive salt entering the transpiration stream will eventually injure cells in the transpiring leaves and further reduce growth [6]. Based on both shoots’ and roots’ fresh and dry weight, the Hail 548 landrace appeared as a salt-tolerant landrace, and both Hail 747 and 1072 landraces were relatively less salt tolerant. Hail 548 shows less reduction in leaf area, plant height, fresh and dry weight of both roots and shoots. The differences in landraces could be the genetic variability among landraces that affect the defense mechanism, for example, by forming an extensive root system to help the plant obtain more water and avoid stress.

### 3.2. The Effect of Salinity on Leaf Pigments, Free Proline, Protein Content, Soluble Sugars

The effect of salinity on physiological pigments was heavily studied in different plant species, including tomato. Chlorophyll pigments were suppressed with an increase in saline conditions [19,28,29]. Wang and Nii [30] observed that a higher chlorophyll content was in plants during salinity stress conditions when expressed on a leaf area basis. However, the content decreased when the plants were transferred to the relief medium, and they concluded that the chlorophyll content in the leaves of salt-stressed plants depended on changes in tissue water content. Jaleel et al. [31] stated that a reduction in chlorophyll content might be because of the increased chlorophyllase activity and the pigment–protein complex’s instability. Delfine et al. [32] reported that a 40% reduction in total chlorophyll in a leaf area basis in 27 d salt-stressed leaves might be because of sodium accumulation, accompanied by a reduction in Ca and Mg uptake, which may have caused the increased permeability of membranes to salt and reduced chlorophyll synthesis. However, Taïbi et al. [33] reported that reducing chlorophyll because of slow synthesis or fast breakdown was a photoprotection mechanism by reducing light absorbance by decreasing chlorophyll contents. This result agreed with that of Elsheery et al. [34]. They concluded that although chlorophyll loss is a negative consequence of stress, it has also been considered an adaptive feature, reducing the light-harvesting and damage to the photosynthetic machinery by activated oxygen radicals under an excess of excitation energy.

Dogan et al. [28] supported the protective role of chlorophyll levels, which may play a key role against stress, and these features can include identifiers for the tolerance of salt. The percentage of reduction or increase in chlorophyll content in landraces investigated in this study showed landraces Hail 747 maintained high chlorophyll content at 150 mM NaCl with values 41.2%, 37%, and 40% for chlorophyll-a, b, and total chlorophyll. In comparison, the Hail 548 landrace showed the highest reductions at 300 mM NaCl with values 48.56%, 47%, and 48%, respectively, for chlorophyll-a, b, and total chlorophyll content. The leaf area of Hail 548 was not affected by salt treatments, but the leaf area was significantly reduced at the highest salinity levels in other landraces. The relative water content exploded in both landraces Hail 548 and 747, while it was unaffected in Hail 1072. The overall chlorophyll content did not significantly differ among genotypes, and treatments suggest that Hail 548 was a relatively salt-tolerant landrace.

The function of carotenoids in photosynthesis is as light energy collectors and quenchers of triplet chlorophyll and single oxygen. They scatter excess energy and can act as powerful chloroplast membrane stabilizers that partition between light-harvesting complexes and the lipid phase of thylakoid membranes, reducing membrane fluidity and lipid susceptibility peroxidation [35,36]. As an antioxidant, carotenoids can detoxify the plants from the effects of ROS. Ziaf et al. stated that carotenoid contents might help differentiate between salt-sensitive and tolerant cultivars. Genotypes with a higher content of carotenoids might quench ROS and be regarded as relatively salt-tolerant [37].

Salinity stress was reported to cause an accumulation of different osmolytes in plant cells and help plants tolerate stress. Some of these osmolytes are proline, protein, and soluble sugars. In this study, there was a significant increase in proline contents in all landraces, and the highest increase was associated with the highest salt level treatment. The response to salt stress treatments was also landrace dependent. In this study, salinity treatment effects were on increasing proline significantly; however, no significant impact was recorded on soluble protein and sugar, reducing the soluble protein and sugar. However, landraces may perform a different tolerant mechanism for salt stress. Hail 747 showed the accumulation of more proline at the highest level of salt stress associated with the high level of relative water content that reduces osmotic stress, while Hail 548 accumulated more proteins and sugars than other landraces at all salt stress levels and performed salinity tolerance mechanisms.

It also reported that the increase in proline percentage is associated with the elevation of salinity in many works [18,19,38]. Plants can adapt to salinity stress through various mechanisms, including osmotic regulation, ion uptake and transport, antioxidant metabolism, hormone metabolism, leaf expansion, stomatal conductance, photosynthesis, and stress signaling [19]. To combat osmotic stress imposed by high salinity, plants should synthesize compatible organic solutes such as proline in the cytosol, affecting both osmo-protection and osmotic adjustment under salinity stress. The accumulation of compatible solutes increases cellular osmolarity, driving an influx of water or reducing the efflux, which provides the turgor necessary for cell expansion. Under osmotic or dehydration stress conditions, membrane integrity must be maintained to prevent protein denaturation. Loukehaich et al. [38] and Soshinkova et al. [39] proposed that compatible solutes such as proline have a valuable role in decreasing the cytoplasmic osmotic potential, facilitating water absorption, and scavenging reactive oxygen species (ROS) molecules. Besides, proline contributes to stabilizing subcellular structures, modulating cell redox homeostasis, supplying energy, and functioning as a signal [40,41]. Marco et al. [42] stated that the accumulation of proline in plants under stress is caused either by the induction of expression of proline biosynthesis genes or by repressing its degradation pathway’s genes.

In tomato, Doganlar et al. [43], and eggplants, Shaheen et al. [44] found a significant reduction in total soluble protein content because of specific stress synthesis proteins. A disruption in the protein synthesis mechanism or increased proteolytic activity was behind reducing protein content in the leaf as salinity increased. These enzymatic activities induce programmed cell death, including nutrient recycling and the selective destruction of misfolded or damaged proteins [45]. Singh et al. [46] suggested that plants under stress accumulate small molecular mass proteins that could be a source of storage nitrogen that could be mobilized after stress relief or removal. These proteins could also have a role in osmotic adjustment [46,47].

The accumulation of soluble carbohydrates has been widely reported as a response to salinity or drought in plants. Loukehaich et al. [38] recorded increasing soluble sugar content for all tomato genotypes under salt stress, and it was more promising in tolerant genotypes. However, Gharbi et al. [15] reported that salt treatment decreased leaf sugar content and lowered total carbohydrate accumulation in the plants compared to controls.

### 3.3. The Effect of Salinity on Some Antioxidant Parameters and MDA Content

Plants under salinity create stress-induced alterations in the cellular metabolism and defense mechanisms to cope with this stress. This includes the over-accumulation of reactive oxygen species (ROS) such as H_2_O_2_ and singlet oxygen O_2_ or hydroxyl radical OH^−1^, inducing oxidative stress. The stresses speed up the damage of most cellular organelle’s components, including protein, lipid, and nucleic acids [20]. These ROS act at lower concentrations as intracellular signaling agents, inducing a positive response in the antioxidant system; however, at high levels, they become toxic and capable of interacting with all kinds of organic molecules, such as nucleic acids and lipids [21]. ROS also influences the expression of several genes. Therefore, it controls many processes such as growth, cell cycle, programmed cell death (PCD), abiotic stress responses, pathogen defense, systemic signaling, and development [48].

The antioxidant system in plants is composed of enzymes such as superoxide dismutase (SOD) and catalase (CAT), and non-enzymatic mechanisms such as ascorbic acid, glutathione, phenolics, and flavonoids, which are involved in the sensing, detoxification, elimination, and neutralization of ROS overproduction [49] to scavenge excess ROS. Superoxide dismutase (SOD) converts superoxide to H_2_O_2_. Hydrogen peroxide is scavenged by catalase (CAT) and different classes of peroxidases. Studies have been conducted to investigate the antioxidants’ role (enzymatic and non-enzymatic stems) in controlling salinity stress in different plant species, including tomato. Parvin et al. [19] found that lipid peroxidation was increased by salt stress, caused by a higher and dose-dependent electrolyte leakage (EL) from both the roots and leaves of tomato plants compared to unstressed plants. Salt stress increased the SOD activity, and a reduction in CAT activity under salinity showed a reduction in the capacity for H_2_O_2_ detoxification, which would cause more considerable oxidative damage. Decreasing lipid peroxidation and membrane damage shows that tomato plants can tolerate salinity and show rapid post-stress recovery by enhancing their antioxidant defense and glyoxalase systems. Ahmad et al. [22] reported an increase in electrolyte leakage, lipid peroxidation, and hydrogen peroxide production in tomato plants subjected to 200 mM NaCl. Salt treatment enhanced antioxidant enzymes such as superoxide dismutase (SOD) and catalase (CAT).

Our results showed that compared with the control treatment, MDA content was increased in both Hail 548 and Hail 747 and reduced in Hail 1072, showing the enhancement of lipid peroxidation and membrane leakage in both landraces of Hail 548 and Hail 747 compared with the Hail 1072 landrace. These results were congruent with [22] findings, which showed exposed *Brassica juncea* to salt stress, leading to increased membrane leakage and the reduced stability of membranes, and enhanced ROS production in salinity-stressed plants trigger the loss of membrane integrity by causing the peroxidation of lipids. Manai et al. [50] found that catalase activity decreased drastically in tomatoes, showing that salinity stress affected the enzyme. Shalata et al. [51] reported that membrane lipid peroxidation gradually increased in the cultivated tomato genotype accompanied by decreased antioxidant enzymes; superoxide dismutase and catalase SOD, CAT activities increased in roots wild relative genotype, in which the level of membrane lipid peroxidation remained unchanged. Mittova et al. [52] explained that the reasons behind the tolerance of wild rather than cultivated tomato to salinity stress were because of

SOD APX (ascorbate peroxidase), and POD (guiacol peroxidase).

SOD (superoxide dismutase), (ascorbate peroxidase), and POD (guiacol peroxidase) activities resolved into several isozymes. While two Cu/ZnSOD isozymes were in cultivated genotype plastids, an additional FeSOD type could also be detected in wild relatives. The selected SOD, APX, and POD isozymes were increased in wild genotypes while decreased in cultivated genotypes. Gharsallah et al. [18] found that CAT activity exploded, and the activity displayed a significant increase within tolerant genotypes concomitant with a pattern of high-antioxidant enzyme activities and displayed differential patterns of gene expression during the response to salt stress. Abdelaal et al. [23] reported upregulated transcriptional levels of the corresponding genes correlated with antioxidant enzymes’ increased activities.

MDA content was increased in both Hail 548 and Hail 747; however, there is a reduction in Hail 1072 with increasing salt stress. The landraces’ responses treated MDA differently. Superoxide dismutase activity (SOD) in Hail 747 and Hail 1072 increased while increasing in catalase (CAT) in Hail 548, showing the antioxidant’s activation defense mechanism. SOD is the first line of defense against ROS, catalyzing the reaction of the dismutation of the superoxide radical to hydrogen peroxide (H_2_O_2_) when the accumulation results in damage to the plant. The elimination of H_2_O_2_ is performed by the enzyme CAT [48]. CAT activity increased in Hail 548, with a maximum increase close to 7.6% when treated with 75 mM NaCl. The increase in CAT activity favors the elimination of H_2_O_2_, both generated by SOD and produced in photorespiration [53]. Abdelaal et al. [23] found that in sweet pepper, an increase in malondialdehyde (MDA) content under the salinity condition and the antioxidants SOD and CAT activities increased salt stress and were higher in the tolerant cultivar. Loukehaich et al. [33] considered a promising tolerant genotype generated less MDA content because osmotic stress damaged the membranes more in sensitive genotypes.

### 3.4. The Effect of Salinity on Some Photosynthetic Parameters and Relative Water Content (RWC)

Photosynthesis is the most fundamental and complicated physiological process that affects plant growth. It involves various components, including photosynthetic pigments and photosystems, the electron transport system, and CO_2_ reduction pathways. Any damage at any level caused by stress may reduce the overall photosynthetic capacity of a green plant [54]. Salinity directly inhibits photosynthesis by closing the stomata, lowering CO_2_ assimilation, obstructing the electron transport chain, and altering the expression of stress-related genes [55].

In this study, photosynthesis was assessed using the net CO_2_ assimilation rate, stomatal conductance, and transpiration rate. Increasing salt concentrations significantly reduced the photosynthesis rate and related traits. On average, the most significant reductions (70.55%) were in the transpiration rate, 7.13% in stomatal conductance, and 72.34% in photosynthetic rate, compared to the control treatment concentrations of 300 mM NaCl. The reduction was also landrace and salt concentration dependent. The reduction in photosynthesis processes was reported in many studies that coincide with our results [16,56,57,58]. Gharbi et al. [15] suggested that stomatal closure is associated with a down-regulation of electron transport rate, which is compensated by an increase in non-photochemical quenching and water reduction potential, and an increase in Na^+^ in vacuoles could decrease cell water potential under saline conditions. Bacha et al. [16] referred to the reduction in net photosynthesis and stomatal conductance, either to stomatal function (the limitation of CO_2_ supply from the partial closure of stomata), or by not a stomatal function (altering the biochemical mechanism of CO_2_ fixation), or both procedures. Gong et al. [58] also reported that the reduction in photosynthetic rate leads to a reduction in plant biomass at salt stress treatments, and the reduction in photosynthesis was mainly because of the lower leaf internal CO_2_ concentration caused by stomatal closure or non-stomatal factors (contents of photosynthetic pigments), biochemical constituents and ultrastructure damage of the chloroplast. They suggested that lower water potential could stimulate the induction of physical or chemical signal materials in the root cause of the reduction in stomatal conductance and transpiration rate. The reduction in the photosynthetic rate under salt stress was believed to be because of ionic and osmotic stresses. Low osmotic potential leads to stomatal closure, and under highly saline conditions, water absorption will be inhibited and then inhibit photosynthesis because of the osmotic effect. Kalaji et al. [17] stated that decreasing photosynthesis in salt-stressed plants was caused by photosystem II damage, and a reduction in the photosystem electron transport rate, the actual quantum yield of photosystem II (PSII), the maximal efficiency of PSII photochemistry were observed in salt-stressed leaves because of the accumulation of Na^+^ and Cl^−^ in mesophyll tissues [59].

The importance of RWC in leaves is not only to maintain leaf turgidity but also to function stomatal conductance. Therefore, a decrease in RWC restricted the photosynthetic rates. Finally, reduced plant growth rate and dry matter accumulation suggest considering RWC in screening genotypes for salinity stress tolerance [37]. The reduction in the dry matter at high salinity was attributed mainly to the reduced leaf area and, to a lesser extent, to a decrease in stomatal conductance in radish [60]. According to pepper results [14], the increased RWC values in salt-tolerant cultivars at lower salt concentrations than salt-sensitive cultivars are because of the accumulation of osmolytes, which makes the surplus of water uptake possible. However, the reduction in RWC at higher concentrations may be attributed to the accumulation of toxic ions such as Na^+^ and Cl^−^, reducing leaf expansion and stomata closure, reducing intercellular CO_2_ partial pressure. Loukehaich et al. [38] explained that the reasons behind the tolerance of genotypes tested to salinity stress were the lower accumulation of Na^+^ and higher relative water content; these salt-tolerant cultivars had a high photosynthetic capacity.

In this study, both Hail 548 and Hail 747 showed higher RWC at a higher level of salt concentrations compared with control, while Hail 1072 showed a reduction in the relative water content at a higher salt level might support the relative tolerance to salinity in Hail 548 and less tolerance in Hail 1072.

### 3.5. The Effect of Salinity on Sodium–Potassium and K^+^/Na^+^ Ratio

The present study results showed that NaCl treatments increased Na^+^ concentration in all landraces, K^+^ decreased in two landraces, and the K^+^/Na^+^ ratio decreased in all genotypes as the salinity level increased. The fact that the sodium concentration increased in response to salt treatment agrees with previous studies in different plant species, including tomato.

The Na^+^ accumulation in plants has many harmful effects; for example, leaf necrosis and reduced shoot and root growth [6]. It decreases the availability of many nutrients because it interferes with K^+^ selective ion channels in the root plasma membrane [61]. Osmotic damage and reduction in the intake of K^+^ ions hinder protein synthesis as it plays a significant role in binding tRNA to ribosomes [62]. Chen et al. [63] stated that the tissue K^+^/Na^+^ ratio has often been named central to salt tolerance in various plant species. A plant’s ability to maintain a high K^+^/Na^+^ ratio, either keeping K^+^ or preventing Na^+^ from accumulating in leaves, is a crucial feature of salt tolerance. Na^+^ must be restricted from entering the cytosol by limiting Na^+^ entry or Na^+^ efflux into the vacuole or the apoplast. Therefore, the capacity to maintain a high K^+^/Na^+^ ratio is one of the key elements in plant salinity tolerance [24,25,28,38]. K^+^ favors the osmotic adjustment in normal and low-level stress conditions. However, a reduced supply of K^+^ during elevated salinity may lead to a high accumulation of stress hormones [64] that could decrease stomatal conductance and the activity of photosynthetic enzymes [65]. Percey et al. [66] reported that a reduction in K^+^ efflux in halophytes is linked to reduced H^+^ efflux, saving energy, allowing more resources to be redirected for plant growth. A high Na^+^ concentration can induce K^+^ deficiency, inhibiting enzymes that require K^+^ [38].

The high concentration of Na^+^ in the growing medium will cause cytoplasmic ion toxicity [67] and disturb cellular ion homeostasis, interference in enzyme activities, and oxidative stress [68]. Injurious hyperosmotic and hyper-ionic effects of salinity lead to membrane disorganization because of reactive oxygen species [69]. The salt-tolerant plants cope with less Na^+^ accumulation by controlling the influx and/or efflux from the cytoplasm to the vacuoles and back the growth medium [70]. The roots of salt-tolerant genotypes accumulate more K^+^ than sensitive genotypes. Raza et al. [13] reported that the control level of salinity was efficient in producing a high ratio of K^+^/Na^+^ and more photosynthetic pigments, but at higher salinity, it reduced substantially compared to the control. Siddiky et al. [26] stated that Na^+^ toxicity led to physiological impairments, including the disruption of K^+^ nutrition, water stress, and oxidative cell damage. Hence, plants must maintain a low Na^+^ concentration by preventing Na^+^ uptake, sequestering Na ions in the vacuole, or regulating Na^+^ homeostasis in the cells by a higher K^+^/Na^+^ ratio. It was postulated that the K^+^/Na^+^ ratio shows a positive relationship with salt tolerance in many crop species, and it might be a valid selection criterion for evaluating salt tolerance [26,71].

## 4. Materials and Methods

The seeds of three tomato Hail landraces were provided by the National Plant Genetic Resources (Gene bank) of Saudi Arabia. The experiment was conducted in a glasshouse (17 °C/23 °C night and day temperature), under a long-day photoperiod (16 h light/8 h dark), at the College of Food and Agriculture Sciences, King Saud University, Saudi Arabia. Ten days after emergence, three seedlings were sown in 500 mL pots filled with pure sand irrigated twice a week with tap water supplied with 1/10 strength of Hoagland solution. Three salinity treatments were applied: 75, 150, and 300 Mm, NaCl, and control (tap water). Four replicates of the split-plot arrangement in a completely randomized design were applied. Genotypes were assigned the main plots, and salinity treatments were in subplots. Plants were harvested and tested for their response to salinity at the seventh fully expanded leaf stage.

Plant height, and the fresh and dry weight of both shoots and roots were measured and recorded, and the leaf area measured using (Meter Cid, Inc Ci- 202 Area). The relative water content was determined according to [72] using the fifth fully expanded leaf using the Equation:(1)$$\mathrm{RWC}=\frac{\left(\mathrm{TFW}-\mathrm{BW}\right)-\mathrm{DW}}{\mathrm{TW}-\mathrm{DW}}\;\times \;100$$

FW is the total fresh weight, BW is the plastic bag weight, TW is the turgid weight, and DW is the leaf’s dry weight. The contents of chlorophyll a (Chl-a), chlorophyll b (Chl-b), total chlorophyll, and carotenoids are determined according to [73] and spectrophotometrically measured using (Uv/Visible Spectrophotometer-LKB-Biochrom-4050) according to [74].

Chlorophyll (a) = {(12.7 × O. D 663) − (2.69 × O. D 645)} × V/1000 × W

Chlorophyll (b) = {(22.9 × O.D 645) − (4.68 × O.D 663)} × V/1000 × W

Total chlorophyll = {(20.2 × O.D 645) + (8.02 × O.D 663)} × V/1000 × W

Carotenoids (Car) = {O.D 480 + (0.114 × O.D 663)} − (0.638 × O.D 645)

O.D: The optical density of the extract at the wavelength shown.

V: volume of extract (mL). W: the weight of fresh leaves (g).

The proline contents were determined according to the method [75] and the standard proline curve (mg/g wet weight) were used to estimate the amount of proline in the wet plant sample. Protein was extracted according to [76] and determined following the [77] protein assay, using bovine serum albumin (BSA) as our protein standard. The sugars were extracted from plant tissues by 80% ethanol [78], and the dissolved sugars were estimated using the [79] method using the standard glucose curve (mg/g dry weight).

The malondialdehyde (MDA) content was determined according to [80] using a UV/Visible Spectrophotometer (LKB-Biochrom 4050, Phoenix, AZ, USA) with the absorbance at 532 nm, 600 nm measured. The method described by Mukherjee and Choudhuri [81] was used to extract the enzymes in 50 mmol potassium phosphate regulator (KH2PO4/K2HPO4 pH = 6.8). In all enzymes, planks were distilled water, and four replicates were used for each treatment. The determination of superoxide dismutase activity (SOD) was estimated using the [82] method. The modification absorption reading was at a 450 nm wavelength using a Micro-Plate Reader (Anthos 2010) and expressed in unit/mg protein (Un/mg protein). The catalase activity (CAT) was determined according to [83], and the absorption was measured at a wavelength of 250 nm using an optical spectrum (Uv/Visible Spectrophotometer—LKB-Biochrom 4050) and expressed in units/mg of protein (Un/mg Protein). The photosynthetic rate, intercellular CO_2_ concentration, conductance to H2O, leaf temperature, and transpiration rate were determined using an LI-6400 photosynthesis system (Li-Cor, Inc., Lincoln, NE, USA) equipped with a standard two × 3 cm leaves corvette and a Li-Cor LI-6400-02B light source used for gas exchange measurements. The ionic contents Na^+^, K^+^, and K^+^/Na^+^ ratio in the dried leaves were determined according to [84], and the concentrations of sodium (Na^+^), potassium (K^+^) were analyzed by a Flame Atomic Absorption Photometer (model 1382).

### Statistical Analysis

The effects of NaCl, genotypes, and the interaction between them were evaluated at a level of *p* < 0.05. Physiological and morphological data collected were subjected to analysis of variance (ANOVA), and the honestly significant difference (HSD) at *p* < 0.05 probability level using Tukey’s test was used to compare the differences among treatment means using [85]. The relative change (increasing or decreasing) in the effect of treatments and genotypes was calculated according to the control treatment.

## 5. Conclusions

The response to salt stress in landraces involved some modifications in morphology, physiology, and metabolism. The Hail 548 landrace may have better protection against salt stress and observed protection against reactive oxygen species (ROS) by increasing enzymatic antioxidants’ activity under salt stress. Further molecular studies regarding salinity stress tolerance to determine key pathways controlling salinity tolerance in tomato landraces have to be conducted.

## Figures and Tables

**Table 1 plants-10-00696-t001:** Mean values of growth-related traits for three Hail landraces under four NaCl stress treatments. Values are the mean of four replications. According to Tukey’s test, different letters within the same columns show significant differences (*p* < 0.05). The higher cases are the differences among genotypes, and the lower cases are the differences among treatments’ mean. ns: means not significant.

Genotype	Treatment	Leaf Area (cm^2^)	Plant Height (cm)	Shoot Fresh Weight (g)	Shoot Dry Weight (g)	Root Fresh Weight (g)	Root Dry Weight (g)
Hail 548	Control	286.92	56.00	241.80	30.23	9.33	1.95
	75 mM	280.15	52.00	145.42	14.58	4.35	0.88
	150 mM	283.74	54.50	169.63	15.78	2.85	0.75
	300 mM	281.29	44.75	110.16	11.70	2.10	0.65
Genotype mean		283.02A	51.81A	166.75A	18.07A	4.66A	1.06A
Hail 747	Control	275.44	55.75	183.90	20.58	6.20	1.45
	75 mM	237.53	49.75	96.21	9.40	2.15	0.65
	150 mM	197.33	44.25	85.06	8.83	1.20	0.38
	300 mM	152.77	32.75	36.53	5.25	0.58	0.13
Genotype mean		215.77B	45.63B	100.43B	11.01B	2.53B	0.65B
Hail 1072	Control	245.37	39.75	181.19	18.85	6.55	0.78
	75 mM	210.60	39.25	140.19	11.80	3.15	0.48
	150 mM	210.83	38.50	85.96	9.95	1.10	0.20
	300 mM	134.27	23.25	30.45	2.98	0.40	0.10
Genotype mean		200.27B	35.19C	109.45B	10.89B	2.80B	0.39B
Treatments mean							
	Control	269.24a	50.50a	202.29a	23.22a	7.36a	1.39a
	75 mM	242.76b	47.00ab	127.28b	11.93b	3.22b	0.67b
	150 mM	230.63b	45.75b	113.55b	11.52b	1.72b	0.44bc
	300 mM	189.44c	33.58c	59.05c	6.64c	1.03b	0.29c
An honestly significant difference (HSD) at *p* < 0.05 probability level using Tukey’s test for:							
Genotypes		15.92	2.05	36.50	3.22	1.25	0.28
Treatment		17.39	3.72	30.10	2.72	1.35	0.23
Genotypes × treatments		30.30	ns	ns	ns	ns	ns

**Table 2 plants-10-00696-t002:** Mean values of chlorophyll-a, chlorophyll-b, total chlorophyll, and carotenoid for three Hail landraces under four NaCl stress treatments. Values are the mean of four replications. According to Tukey’s test, different letters within the same columns indicate significant differences (*p* < 0.05). The higher cases are the differences among genotypes, and the lower cases are the differences among treatments’ mean. ns: means not significant. Chl_a: chlorophyll-a; Chl_b: chlorophyll-b; T_chlo: total chlorophyll.

Genotype	Treatment	Chl_a (mg/g Fresh Weight)	Chl_b (mg/g Fresh Weight)	T_chlo (mg/g Fresh Weight)	Carotene (mg/g Fresh Weight)
Hail 548	control	0.072	0.028	0.101	0.508
	75 mM	0.061	0.026	0.086	0.498
	150 mM	0.066	0.026	0.093	0.482
	300 mM	0.037	0.015	0.052	0.311
Genotype mean		0.059A	0.024A	0.083A	0.450A
Hail 747	control	0.058	0.023	0.080	0.465
	75 mM	0.074	0.029	0.103	0.674
	150 mM	0.082	0.031	0.113	0.657
	300 mM	0.066	0.025	0.091	0.584
Genotype mean		0.070A	0.027A	0.097A	0.595A
Hail 1072	control	0.113	0.042	0.155	0.711
	75 mM	0.095	0.034	0.128	0.664
	150 mM	0.073	0.027	0.099	0.549
	300 mM	0.089	0.031	0.120	0.600
Genotype mean		0.092A	0.033A	0.126A	0.631A
Treatments mean					
	control	0.081a	0.031a	0.112a	0.561a
	75 mM	0.077a	0.029a	0.106a	0.612a
	150 mM	0.074a	0.028a	0.102a	0.563a
	300 mM	0.064a	0.024a	0.088a	0.498a
An honestly significant difference (HSD) at *p* < 0.05 probability level using Tukey’s test for:					
Genotypes		ns	ns	ns	ns
Treatment		ns	ns	ns	ns
Genotypes × treatments		0.04	ns	ns	ns

**Table 3 plants-10-00696-t003:** Mean values of proline, protein, and soluble sugars for three Hail landraces under four NaCl stress treatments. Values are the mean of four replications. According to Tukey’s test, different letters within the same columns show significant differences (*p* < 0.05). The higher cases are the differences among genotypes, and the lower cases are the differences among treatments’ mean. ns: means not significant.

Genotype	Treatment	Proline (mg/g Fresh Weight)	Protein (mg/g Fresh Weight)	Soluble Sugar (mg/g Dry Weight)
Hail 548	control	2.224	1.386	16.717
	75 mM	10.127	1.348	14.167
	150 mM	37.694	1.171	11.686
	300 mM	42.636	1.271	12.734
Genotype mean		23.176B	1.294A	13.826A
Hail 747	control	1.439	0.474	12.013
	75 mM	19.323	0.451	12.768
	150 mM	25.525	0.470	11.319
	300 mM	90.045	0.469	11.581
Genotype mean		34.100A	0.466B	11.920B
Hail 1072	control	3.930	0.477	8.401
	75 mM	25.185	0.476	10.467
	150 mM	25.061	0.456	9.697
	300 mM	69.449	0.505	9.248
Genotype mean		30.900AB	0.478B	9.453C
Treatments mean				
	control	2.531c	0.779a	12.377a
	75 mM	18.212b	0.760a	12.467a
	150 mM	29.427b	0.700a	10.901a
	300 mM	67.376a	0.748a	11.188a
An honestly significant difference (HSD) at *p* < 0.05 probability level using Tukey’s test for:				
Genotypes		10.39	0.593	1.10
Treatment		12.13	ns	ns
Genotypes × treatments		20.90	0.628	ns

**Table 4 plants-10-00696-t004:** Mean values of superoxide dismutase (SOD), catalase activity (CAT), and malondialdehyde (MDA) traits for three Hail landraces under four NaCl stress treatments. Values are the mean of four replications. According to Tukey’s test, different letters within the same columns show significant differences (*p* < 0.05). The higher cases are the differences among genotypes, and the lower cases are the differences among treatments’ mean. ns: means not significant.

Genotype	Treatment	SOD (Un/mg Protein)	CAT (Un/mg Protein)	MDA (µM/g Fresh Weight)
Hail 548	control	1.357	0.079	17.38
	75 mM	0.678	0.085	26.505
	150 mM	0.553	0.074	19.995
	300 mM	0.472	0.083	15.81
Genotype mean		0.765A	0.080	19.298
Hail 747	control	0.522	0.101	17.515
	75 mM	0.490	0.046	16.915
	150 mM	0.531	0.033	23.91
	300 mM	0.545	0.072	22.785
Genotype mean		0.522B	0.063	20.281
Hail 1072	control	0.405	0.080	18.135
	75 mM	0.403	0.076	17.36
	150 mM	0.427	0.070	15.655
	300 mM	0.423	0.050	11.16
Genotype mean		0.414B	0.069	15.578
Treatments mean				
	control	0.761a	0.087	17.677
	75 mM	0.524b	0.069	20.260
	150 mM	0.504b	0.059	19.853
	300 mM	0.480b	0.068	16.585
An honestly significant difference (HSD) at *p* < 0.05 probability level using Tukey’s test for:				
Genotypes		0.13	ns	ns
Treatment		0.07	ns	ns
Genotypes × treatments		0.16	ns	ns

**Table 5 plants-10-00696-t005:** Transpiration rate, stomatal conductance, photosynthetic rate, and relative water content (RWC) for three Hail landraces under four NaCl stress treatments. Values are the mean of four replications. According to Tukey’s test, different letters within the same columns indicate significant differences (*p* < 0.05). The higher cases are the differences among genotypes, and the lower cases are the differences among treatments’ mean. ns: means not significant.

Genotype	Treatment	Transpiration Rate (mmol H_2_O m^−2^ s^−1^)	Stomatal Conductance (mol H_2_O m^−2^ s^−1^)	Photosynthetic Rate (µmol CO_2_ m^−2^ s^−1^)	RWC%
Hail 548	control	2.014	1.445	17.627	0.72
	75 mM	0.346	1.350	12.983	0.90
	150 mM	0.393	1.355	9.165	0.80
	300 mM	0.176	1.342	9.389	0.52
Genotype mean		0.732A	1.373B	12.291B	0.73A
Hail 747	control	1.192	1.417	21.473	0.79
	75 mM	0.446	1.363	12.395	0.69
	150 mM	1.135	1.350	10.81	0.69
	300 mM	0.304	1.355	6.258	0.48
Genotype mean		0.769A	1.391B	12.73AB	0.66A
Hail 1072	control	1.493	1.487	22.861	0.15
	75 mM	0.604	1.390	15.832	0.36
	150 mM	0.661	1.404	10.222	0.36
	300 mM	0.808	1.421	6.324	0.20
Genotype mean		0.892A	1.426A	13.810A	0.27B
Treatments mean					
	control	1.567a	1.450a	20.654a	0.56a
	75 mM	0.465c	1.368c	13.737b	0.65a
	150 mM	0.730b	1.396b	10.066c	0.61a
	300 mM	0.429c	1.373c	7.324d	0.40b
An honestly significant difference (HSD) at *p* < 0.05 probability level using Tukey’s test for:					
Genotypes		ns	0.025	1.440	0.12
Treatment		0.220	0.020	2.163	0.12
Genotypes × treatments		0.444	0.040	3.534	0.28

**Table 6 plants-10-00696-t006:** Content of sodium (Na), potassium (K), and K/Na ratio for three Hail landraces under four NaCl stress treatments. Values are the mean of four replications. According to Tukey’s test, different letters within the same columns show significant differences (*p* < 0.05). The higher cases are the differences among genotypes, and the lower cases are the differences among treatments’ mean. ns: means not significant.

Genotype	Treatment	Na^+^ (mg/g)	K^+^ (mg/g)	K^+^/Na^+^
Hail 548	control	4.71	62.02	13.20
	75 mM	10.92	75.32	6.90
	150 mM	33.51	84.62	2.50
	300 mM	51.28	86.58	1.70
Genotype mean		25.11A	77.13 B	6.07A
Hail 747	control	6.46	130.94	20.30
	75 mM	23.62	149.57	6.30
	150 mM	43.53	100.38	2.30
	300 mM	43.38	85.27	2.00
Genotype mean		29.25A	116.54A	8.59A
Hail 1072	control	14.96	128.9	8.60
	75 mM	19.53	106.04	5.40
	150 mM	30.81	98.59	3.20
	300 mM	29.99	80.59	2.70
Genotype mean		23.82A	103.53A	5.00A
Treatments mean				
	control	8.71b	107.29a	14.00a
	75 mM	18.02b	110.31a	6.20 b
	150 mM	35.95a	94.53 a	2.70c
	300 mM	41.55a	84.15a	2.15c
An honestly significant difference (HSD) at *p* < 0.05 probability level using Tukey’s test for:				
Genotypes		ns	32.16	ns
Treatment		19.39	ns	3.37
Genotypes × treatments		ns	ns	ns

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
