# Peer review of "Assessment of Morpho-Physiological, Biochemical and Antioxidant Responses of Tomato Landraces to Salinity Stress"

_plants, 2021, doi:10.3390/plants10040696_

Round 1
Reviewer 1 Report
- To revise 'proline content' related to new tomatoes landraces stress response to salinity. In the Abstract : line 16- is general statement; re-phrase and explain role of proline increasing for the investigated genotypes.
- Discussion in some places is given for adult plants /at fruits stage. Literature overview -to relate to seed germination and seedling ontogenetic stages since authors made investigation at early plant growth.
Author Response
- To revise 'proline content' related to new tomatoes landraces stress response to salinity. In the Abstract: line 16- is general statement; re-phrase and explain role of proline increasing for the investigated genotypes.
In the abstract, we add the statement “the 300 mM NaCl treatment recorded the most proline content increases (67.37 mg/g fresh weight), with a percent increase in proline reached 61.67% in Hail 747”.
- Discussion in some places is given for adult plants /at the fruits stage. Literature overview -to relate to seed germination and seedling ontogenetic stages since authors made investigation at early plant growth.
Thank you, the paragraph mentioned for adult plants/at fruit stage is deleted
Reviewer 2 Report
Dear Authors,
Please check the sentence: `The response of tomato growth to salinity stress is genotypes and concentration-dependent.` (rows 174-175).
Please avoid throughout the manuscript - to start a sentence with `[7]...` or `However, [7]...`. Use instead the name of author/s and indicate then or at the end the number of reference between [].
Row 266 - use `osmo-protection` instead of `Osmo-protection`.
Please make uniform throughout the manuscript - use the numbers in subscript in chemical formulas - CO2, H2O2, O2..
Author Response
- Please check the sentence: `The response of tomato growth to salinity stress is genotypesand concentration-dependent. ` (rows 174-175).
Thank you, we meant that genetic makeup of the genotype, the microenvironment (salt treatments in this study), and interaction effects are responsible for variation in growth among tomato
- Please avoid throughout the manuscript - to start a sentence with `[7]...` or `However, [7]...`. Use instead the name of author/s and indicate then or at the end the number of reference between [].
Thank you, we corrected all cases.
- Row 266 - use `osmo-protection` instead of `Osmo-protection`.
Thank you, the term is changed,
- Please make uniform throughout the manuscript - use the numbers in subscript in chemical formulas - CO2, H2O2, O2..
Thank you, we considered the comment, and all terms are changed
Reviewer 3 Report
The paper is focused on the assessment of Morpho-physiological, biochemical and antioxidant responses of tomato landraces to salinity stress. It is an interesting field of research but some changes must be performed before acceptance.
First of all, the introduction appear very scarce in relation at the discussion section in which part of the sentence could be transpose to the introduction.
The Authors perform statistic related to different parameters and the interaction between treatment and Cv is not significant. Because there is no interaction the letters cannot be inserted in the table and consecutively, there are not differences among cv.
Authors declare that: Similar letters signify no significant differences whereas different letters signify significant differences at 0.05 level of significance by Tukey test. In the table is reported LSD test! Please indicate the correct post- test used and the significance of the test.
Since in the current forms statistical analyses reveal no difference among tomato cv, I suggest analysing data in relation to cv and each treatment to understand the best one in relation to Na concentration.
All data in the manuscript are reported without SD. Please introduce these values.
Some key words are present also in in title please change with news one such as ROS, enzyme activity, growth…
Line 64 typing error such as 300mM instead 300 mM, similar Line 70 1.96 % instead of 1.95% etc…please verify in the text
Please re-calculate the RWC is a % of wight; is impossible that was more than 100!!
Please verify the Na concentration in the organs.
Author Response
- The Authors perform statistic related to different parameters and the interaction between treatment and Cv is not significant. Because there is no interaction the letters cannot be inserted in the table and consecutively, there are not differences among cv.
Thank you, Since HSD value presented in the table, there is no need to show letters in the table. So, we drop the letter from the tables.
- Authors declare that: Similar letters signify no significant differences whereas different letters signify significant differences at 0.05 level of significance by Tukey test. In the table is reported LSD test! Please indicate the correct post- test used and the significance of the test.
Thank you, Tukey test was performed and Honestly significant difference (HSD) at P< 0.05 probability level using Tukey’s test for each factor was fixed in each table.
- Since in the current forms statistical analyses reveal no difference among tomato cv, I suggest analyzing data in relation to cv and each treatment to understand the best one in relation to Na concentration.
Thank you, however, the experiment is laid out using split-plot in RCBD. We tested the design efficiency and showed that split plot design was more efficient than RCBD on comparison with sub-plots.
- Some key words are present also in in title please change with news one such as ROS, enzyme activity, growth
Thank you, keywords changed
- Line 64 typing error such as 300mM instead 300 mM, similar Line 70 1.96 % instead of 1.95% etc…please verify in the text
Thank you all typing errors are corrected
- Please re-calculate the RWC is a % of wight; is impossible that was more than 100!!
Thank you very much, we have corrected the values in this trait and reanalyzed the data
- Please verify the Na concentration in the organs
Thank you, in material and method the dried leaves were used in the analysis
Round 2
Reviewer 3 Report
The manuscript has been improved as suggest. From my point of view it can be published in the current form.
Author Response
Dear reviwer
Thank you very much for your valuable comments which improved our work and become publisher in this respective journal. Best regards